# Bergamot Essential Oil: A Method for Introducing It in Solid Dosage Forms

**DOI:** 10.3390/foods11233860

**Published:** 2022-11-29

**Authors:** Ylenia Zambito, Anna Maria Piras, Angela Fabiano

**Affiliations:** 1Department of Pharmacy, University of Pisa, Via Bonanno 33, 56126 Pisa, Italy; 2Research Centre for Nutraceutical and Healthy Foods “NUTRAFOOD”, University of Pisa, Via del Borghetto 80, 56124 Pisa, Italy

**Keywords:** bergamot essential oil, quaternary ammonium chitosan derivative, methyl-β-cyclodextrin, conjugate chitosan derivative/cyclodextrin, polyphenol protection

## Abstract

Bergamot essential oil (BEO) possess antimicrobial, antiproliferative, anti-inflammatory, analgesic, neuroprotective, and cardiovascular effects. However, it is rich in volatile compounds, e.g., limonene, that are susceptible to conversion and degradation reactions. The aim of this communication was to prepare a conjugate based on a quaternary ammonium chitosan derivative (QA-Ch) and methyl-βCD (MCD), coded as BEO/QA-Ch-MCD, to encapsulate BEO in order to stabilize its volatile compounds, eliminate its unpleasant taste, and convert the oil in a solid dosage form. The obtained conjugate, BEO/QA-Ch-MCD, was highly soluble and had a percentage of extract association efficiency (AE %), in terms of polyphenols and limonene contents, of 22.0 ± 0.9 and 21.9 ± 1.2, respectively. Moreover, stability studies under UV stress in simulated gastric fluid showed that BEO/QA-Ch-MCD was more able to protect polyphenols and limonene from degradation compared to free BEO or BEO complexed with MCD (BEO/MCD). The complexation and subsequent lyophilization allowed the transformation of a liquid into a solid dosage form capable of eliminating the unpleasant taste of the orally administered oil and rendering the solid suitable to produce powders, granules, tablets, etc. These solid oral dosage forms, as they come into contact with physiological fluids, could generate nanosized agglomerates able to increase the stability of their active contents and, consequently, their bioavailability.

## 1. Introduction

In recent years, essential oils derived from plants have raised great interest due to their numerous healthy properties. Among them, bergamot essential oil (BEO), which is extracted from the peel of *Citrus bergamia*, has antimicrobial, antiproliferative, anti-inflammatory, analgesic, neuroprotective, and cardiovascular effects [1,2]. Moreover, BEO has an intense fragrance and aroma, and for these reasons is widely used not only in the pharmaceutical industry, but also in the cosmetic and food industry [3]. BEO is rich in volatile compounds, such as limonene and linalool, susceptible to conversion and degradation reactions [4]. Indeed, essential oils are highly volatile, and their use is rather limited because they are photo- and thermosensitive, and have a very low water solubility. These limitations restrict the beneficial effects of essential oils; thus, the development of an efficient delivery system for them represents a challenging task. Along with traditional dosage forms such as ointments, solutions, and gels, modern technologies have been proposed for the controlled release of essential oils [5], e.g., it has been demonstrated that essential oils could be protected from degradation by encapsulating them in polymeric particles [6] or cyclodextrins. Microencapsulation has been frequently used to incorporate liquid substances into solid carriers, however, microparticles are generally obtained by a spray drying technique, and it is important to set up the drying conditions (feeding rate and the air inlet and outlet temperatures) to obtain a finished product able to retain the encapsulated material and enhance the stability of volatile compounds [7]. Nanoparticles are subcellular in size and have a large surface area available to interact with the biological support. Due to these characteristics, nanoparticles offer promising means of improving the bioavailability of nutraceutical compounds in water-rich phases or liquid–solid interfaces. Moreover, nanoparticles could act as reservoir systems, controlling the release of bioactives [8]. Depending on the nature of the encapsulated bioactive, it may be not worth developing the often complicated nanoparticle dispersions, especially when they weakly interact with the bioactive [9]. A more promising handling approach than those cited above is the use of cyclodextrins (CDs). CDs are cyclic oligosaccharides obtained from the enzymatic degradation of starch. They are non-toxic, and do not interfere with the biological properties of the guest. In addition, CDs are relatively cheap biodegradable materials, and encapsulation can be performed both in solution and in the solid state [10]. It has been reported that inclusion complexes (ICs) between CDs and essential oils can overcome such limitations of essential oils as poor water solubility, unpleasant odor, volatility, etc. In addition, these ICs have good stability when heated during industrial food processing and during storage for a longer period than the essential oils themselves [11]. α-, β-, and γCDs have been approved by FDA as food additives. βCDs have a low solubility in water; the reason for this is that βCD derivatives, such as 2-hydroxypropyl-βCD (HPβCD) and methyl-βCD (MCD), are preferred for the preparation of aqueous pharmaceutical solutions [12]. In order to enhance the beneficial outcomes of oral intake of essential oils, the complexation with CDs has been combined with the mucoadhesive and absorption enhancer properties of chitosan [13]. In particular, a quaternary ammonium chitosan derivative (QA-Ch) was coupled with MCD, leading to a conjugate (QA-Ch-MCD) with high complexing ability, mucoadhesive characteristics, and good cytocompatibility [14]. This conjugate was also used to prepare nanosized carrier systems for the oral administration of dexamethasone [15], and it was demonstrated that using the simple macromolecular complex was more advantageous than using nanosized carriers. The present short communication reports the use of the conjugate QA-Ch-MCD to encapsulate BEO in order to stabilize the volatile bioactives of the oil against degradation reactions, eliminate its unpleasant taste upon oral administration, and convert the oil into a solid, to improve its physical and/or chemical stability.

## 2. Materials and Methods

### 2.1. Materials

Folin–Ciocalteu reagent, gallic acid, limonene standard solution, 1-ethyl-3-(3-dimethylaminopropyil) carbodiimide hydrochloride, cellulose membrane tubing MW cut-off 12.5 kDa, low-molecular-weight chitosan, dimethyl sulfoxide (DMSO), 1,6-hexamethylene diisocyanate (HMDI), and triethylamine (TEA) were purchased from Merck (Darmstadt, Germany). BEO was from Alidans (Pisa, Italy). QA-Ch was synthesized according to [16] and conjugated with MCD as already described [17]. The resulting QA-Ch-MCD had a molecular weight of 603 kg/mol and the following features: 8.8% acetylation degree, 33.1% degree of quaternarization with *n* = 4 (diethyldimethylene ammonium) length pendant chains, and 45.5% MCD degree of functionalization. All aqueous solutions/dispersions were prepared in deionized water.

### 2.2. Determination of Total Polyphenols and Limonene Content in BEO

BEO was analyzed for total content in polyphenols (TPC) by the Folin–Ciocalteu colorimetric method [18], and for limonene content by High Performance Liquid Chromatography (HPLC) using an Aeris 3.6 μm, PEPTIDE XB-C18 Å, 250 × 4.6 mm column, a mobile phase (flow rate 1 mL/min) of methanol/water (85:15) and UV detection set at 210 nm.

### 2.3. Preparation of the Complexes BEO/MCD and BEO/QA-MCD

BEO was complexed with MCD through inclusion complex formation following the procedure reported by Rakmai et al., 2017 [19], slightly modified. Briefly, 10 mg of MCD (or 15 mg of QA-Ch-MCD) was dissolved in 5 mL of deionized water under stirring at room temperature for 1 h. Subsequently, BEO (200 μL) was slowly added to the MCD aqueous solution, and the resulting mixture was stirred at room temperature and protected from light for 24 h. Following centrifugation (Termo Scientific MTX 150, Darmstadt, Germany), the non-encapsulated BEO was discarded, and the aqueous solution was lyophilized (VirTis adVantage-53, Stereoglass, Perugia, Italy). The lyophilized product was redispersed in water and characterized for percentage of extract association efficiency (AE %), in terms of polyphenols and limonene contents. Both were determined by direct solubilization of 1 mg of complex in 100 μL of methanol and analyzing the solution for polyphenols by the Folin–Ciocalteu method, or for limonene by HPLC. AE % was calculated as follows (Equation (1)):(1)$$\mathrm{AE}\;\%\;=\;\left(M_{f}/M_{t}\right)\times 100$$
where M_f_ is the mass of polyphenols or limonene found in the lyophilized product and M_t_ is the total mass of polyphenols or limonene used to prepare the complex.

The obtained products were analyzed in triplicate in water at 25 °C, for size and ζ-potential, by dynamic light scattering (DLS, Malvern Zetasizer Nano ZS).

### 2.4. Spectroscopic Characterization of the Complex

#### 2.4.1. Attenuated Total Reflectance-Fourier Transform InfraRed (ATR-FTIR) Analysis

ATR-FTIR spectra of pure BEO, QA-Ch-MCD and the physical mixture BEO/QA-Ch/MCD (ratio 1:1, *w*/*w*) were collected at room temperature in the wavenumber range 400–4000 cm^−1^ with a resolution of 2 cm^−1^ and 128 scans using a Cary 660 series FTIR (Agilent Technologies, Milan, Italy). The acquired spectra were processed (WaveMetrics, Inc., SW Nimbus, Portland, OR, USA) software. Spectrum subtraction was performed using the 1738 cm^−1^ band as a reference.

#### 2.4.2. UV-Visible Analysis

UV-visible measurements were carried out using a Lambda 25 Perkin Elmer spectrometer, in the wavelength region from 200 to 700 nm, at room temperature. The aqueous tested samples were BEO (1.7 mg/mL), QA-Ch-MCD (2 mg/mL), and 0.2 mg/mL of BEO/QA-Ch-MCD. This latter lower concentration was adopted to limit the scattering of the BEO/QA-Ch-MCD aggregates in water. The acquired spectra were processed using IGOR pro 9.01 (WaveMetrics, Inc.) software.

### 2.5. Polyphenols and Limonene Stability under UV Irradiation

The stability of polyphenols and limonene contained in BEO, free or complexed with MCD or QA-Ch-MCD, was evaluated under UV stress (254 nm, 2.1 mW/cm^2^), in simulated gastric fluid (SGF) pH 1.2 (500 mL, 40 g HCl 1N and 1 g NaCl). In detail, 5 mg of BEO/MCD or BEO/QA-Ch-MCD containing 1.75 mg of BEO were dissolved in 200 μL of SGF under UV irradiation and continuous stirring. At 30 min intervals over 4 h, samples were withdrawn and analyzed for TPC and limonene content. For comparison, the same procedure was used for TPC and limonene content in free BEO (2 μL corresponding to 1.75 mg of BEO). 

### 2.6. Statistical Data Treatment

Experiments were replicated (*n* = 3–6), and the results were averaged and the statistical differences between means was assessed by the ordinary one-way ANOVA test. Differences were considered significant for *p* values lower than 0.05. 

## 3. Results 

### 3.1. Determination of TPC and Limonene Content in BEO

Although reference is made to TPC, it should be noted that the Folin–Ciocalteu reagent is not specific for phenolic compounds, and can also react with other molecules with reducing characteristics, as presented in a previous extract [20]. BEO was characterized by a TPC of 0.667 mg/g of oil and a limonene content of 417.7 mg/g, in line with the values reported in the literature [21,22]. Additionally, the data concerning limonene are in line with the manufacturer data sheet.

### 3.2. BEO/MCD and BEO/QA-MCD Complexes

In order to fine-tune the best complexation conditions, incubation time as well as BEO/MCD ratio were varied, and the best conditions were selected according to the highest achieved AE % (Table 1). AE % values were not influenced by the time of agitation (24 h or 48 h); similarly, by increasing the amount of BEO in the formulation only a slight decrease in AE % was observed. According to the obtained results, the conditions of run 1 were selected and applied for the complexation with the conjugate polymer QA-Ch-MCD.

The AE % values for polyphenols and limonene contents in the BEO/QA-Ch-MCD conjugate were 22.0 ± 0.9 and 21.9 ± 1.2, respectively, not much different from the values for BEO/MCD. 

Size, polydispersity index (PdI), and ζ-potential values are reported in Table 2. The DLS analysis of the BEO/QA-Ch-MCD conjugate showed that the conjugation process led to agglomerates with nanometer size, not different from those formed from BEO/MCD. However, the agglomerates obtained from BEO/QA-Ch-MCD had a more inhomogeneous distribution than those from BEO/MCD, as demonstrated by the PdI values, which confirms the ability of CD host/guest complexes to self-assemble in water [23,24]. In both cases, the ζ-potential absolute values indicate that the agglomerates were reasonably stable, leading to a colloidal dispersion of the lipophilic compounds [25]. If we compare the ζ-potential values obtained with the two agglomerates under study, positive in the case of BEO/QA-Ch-MCD and negative in that of BEO/MCD, we can observe how much the fixed positive charge present on QA-Ch affects the ζ-potential values of the relevant agglomerate [26].

### 3.3. BEO/QA-Ch-MCD Interaction

Spectroscopic methods are mainly applied for the evaluation of guest/host interaction, evidencing the spectral differences deriving from the variation of the chemical environment surrounding the complexed guest molecule [17,27]. ATR-FTIR and UV-Vis spectroscopies were applied (Figure 1). Concerning the ATR-FTIR spectra, for BEO, the most abundant characteristic absorptions were present at 3500 cm^−1^ (O-H stretch), 3100–2760 cm^−1^ (C-H stretch of the aliphatic portion), ~1738 cm^−1^ (C=O stretch of fatty acid esters), 1240 cm^−1^ (C-N), and at 1020, 884, and 918 cm^−1^, attributed to terpenoid components present in essential oils [28]. The QA-Ch-MCD spectrum showed bands in the 3625–3100 cm^−1^ (OH and NH stretching) range, the alkyl stretching C-H bands were found in the 3000–2850 cm^−1^ range, whereas the carbamate and ureic characteristic bands occurred at 1704 and 1618 cm^−1^ (stretching C=O), at 1537 and 1580 cm^−1^ (bending N-H), and the prominent saccharide band at 1020 cm^−1^. The FTIR spectrum of BEO/QA-Ch-MCD was essentially an average of the QA-Ch-MCD and BEO spectra. The subtraction of the Qa-Ch-MCD spectrum from that of the complex (green line in the box of Figure 1a) results in it being overimposable to that of BEO, as confirmed by the ester C=O stretching at 1738 cm^−1^. However, the aliphatic portion presents a shift of the main BEO peak: despite the fact that the bands at 2966 and 2857 cm^−1^ are perfectly recovered, the 2918 is shifted to 2922, indicating an interaction between BEO and the cavity of QA-Ch-MCD.

The UV absorption spectra of BEO (Figure 1b) consist of two main bands centered at 250 and 320 nm that could be attributed to π, π* electronic transitions of diene, and to π* transition of C=O and C=C groups, respectively [29]. Such bands appear shifted in the BEO/QA-Ch-MCD spectrum, as evidenced in the box of Figure 1b, where the contribution of QA-Ch-MCD is reduced by applying second derivative processing. Both the ATR-FTIR and UV-Vis spectra of BEO are dominated by the bands of its three main components, namely, limonene, linalyl acetate, and linalool. Since the detected shifts are correlated to these components, it is confirmed that there is a complexing interaction with the polymeric cyclodextrin derivative. 

### 3.4. Polyphenols and Limonene Stability under UV Irradiation

Studies of BEO complexed with MCD or with QA-Ch-MCD were conducted in FGS. The stability plots for polyphenols or limonene contained in BEO, free or complexed with MCD or QA-Ch-MCD, are reported in Figure 2a,b. BEO/MCD and BEO/QA-Ch-CD were able to protect polyphenols and limonene from degradation, as demonstrated by stability studies in SGF. In particular, the polyphenols of free BEO degraded completely after 150 min, whereas in the case of BEO/MCD and BEO/QA-Ch-MCD, the undegraded percentage of polyphenols was 15% and 30%, respectively. In the case of limonene, its degradation was faster under the assayed condition, and the study lasted only 2 h. The limonene in pure BEO degraded completely after 20 min, whereas the limonene in the BEO/MCD complex degraded completely after 90 min. QA-Ch-MCD was able to protect limonene for a prolonged length of time; the undegraded percentage of limonene was 15% after 120 min. The results show that only when MCD was covalently bound to QA-Ch was there significant protection of polyphenols and limonene from UV degradation. These results are in agreement with those previously found with the labile dalargin peptide [17]. 

## 4. Discussion

Bergamot has numerous nutraceutical properties correlated to the presence of polyphenols, and it was previously demonstrated that peels contain a higher amount of TPC compared to pulp [18]. Considering that BEO is extracted from the peel of *Citrus bergamia*, our results suggest that BEO could provide the body with levels of polyphenols similar to those derived from the consumption of just one fruit, which weighs 80–200 g. Polyphenols undergo oxidation, especially when they come into contact with gastric fluids. In addition, volatile compounds, such as limonene, can be degraded easily, e.g., by oxidation, heating, light exposure, and volatilization. Given these factors, it is important to protect bioactive compounds of BEO, in our case complexing them with CDs. Chemically modified CDs, such as MCD, enhance the solubility of essential oils and have a good AE (%) [30]. Indeed, the results shown in Table 1 indicate a satisfactory AE (%) of limonene and polyphenols in all cases. However, the results cannot be compared with the data reported in the literature, because several factors may contribute to different values of AE (%), such as the loss of oil, or evaporation during the complexation or drying processes [31]. Another important strategy to enhance the oral absorption of hydrophobic molecules is represented by the use of QA-Ch-MCD. Since QA-Ch-MCD has been shown to be more capable than MCD of protecting the dalargin peptide from proteolytic attack by intestinal enzymes [17], it was thought that this complexing agent was also able to protect the polyphenols and limonene present in BEO from degradation in the gastrointestinal environment. Furthermore, QA-Ch-MCD was found to be highly soluble and a good complexing agent [15,17,32]. The lyophilized BEO/QA-Ch-MCD and BEO/MCD conjugates were freely water-soluble, as expected. Size, polydispersity index (PdI), and ζ-potential values represent the most important parameters in the characterization of inclusion complexes because they determine the ability of the complexes to be used in selected applications [33]. In particular, the size of the inclusion complexes affects not only their characteristics, such as their surface characteristics, their stability, and the release of the bioactive, but also the type of application [34]. Because there is not an optimal size, it is important to measure the polydispersity index (PdI), which represents the size distribution of the inclusion complexes. The ζ-potential values indicate the stability of inclusion complexes in suspension [35]; in fact, an increase in the absolute value of the ζ-potential reflects an increase in the repulsive forces that reduces their tendency to aggregate. The size and PdI values of BEO/MCD and BEO/QA-Ch-MCD obtained were reproducible. BEO/MCD exhibited a mean diameter in the range of 359.70 ± 8.50, and that of BEO/QA-Ch-MCD was 414.20 ± 36.06; meanwhile, the PdI values were 0.33 ± 0.01 and 0.67 ± 0.06, respectively, indicating a more uniform size dispersion of BEO/CD than BEO/QA-Ch-MCD. This difference could be ascribed to a greater tendency of BEO/QA-Ch-MCD to agglomerate. Probably, the agglomeration does not occur uniformly, creating variability in the size of BEO/QA-Ch-MCD. Furthermore, the ζ-potential values were −19.90 ± 0.78 for BEO/MCD and +44.60 ± 5.24 for BEO/QA-Ch-MCD, which are high enough to indicate the formation of stable complexes with low tendencies to aggregate. In the food industry, CDs have been used for controlling flavors, odor masking, and for edible food packaging. The FDA approved the use of UV technology for the treatment of food and beverages. Many polyphenols degrade when exposed to light, high temperatures, oxygen, and certain pH conditions. It is known that light exposure accelerates polyphenol degradation by oxidation processes [36]. Since these could overcome the antioxidant effects of BEO, to speed up the stability tests, the dispersions were irradiated for the entire duration of the experiment with UV-C [37]. Encapsulation of BEO in QA-Ch/MCD has been shown to increase the stability of polyphenols for 240 min, and of limonene for 120 min, whereas in the case of pure BEO, polyphenols and limonene degrade completely after 150 min and 20 min, respectively. CDs probably protect polyphenols because the hydroxyl groups of polyphenols, responsible for their antioxidant activity, are oriented towards the cavity of CDs [38]. Moreover, the results obtained with limonene could be related to the presence of limonene within the CD structure. The ATR-FTIR analysis confirmed the formation of the BEO-CD solid complex. Indeed, changes in the characteristic band of the guest molecules could be the result of the limited stretching vibrations of the guest molecule caused by inclusion in the CD cavity. As already demonstrated with dalargin peptide complexed with QA-Ch-MCD [39], the conjugate was not only more soluble than the corresponding progenitor polymers, and hence exhibited greater handling ability, it was also able to protect the peptide from enzymatic degradation much more effectively than the inclusion complex with pure MCD. Similar results were found with BEO/QA-Ch-MCD, which encourage the use of the QA-Ch-MCD conjugate to stabilize volatile compounds present in essential oils. 

## 5. Conclusions

The conjugate QA-Ch-MCD showed high efficiency in complexing polyphenols and limonene contained in BEO, and protecting them from degradation even in extreme oxidizing conditions. The successful formation of the inclusion complex and encapsulation of BEO was confirmed by ATR-FTIR spectroscopy. The complexation and subsequent lyophilization allowed the transformation of a liquid (BEO) into a solid dosage form that could not only eliminate the unpleasant taste of the orally administered bergamot oil, but also render the solid material suitable for producing powders, granules, tablets, etc. These solid oral dosage forms, as they come into contact with physiological fluids, could generate nanosized agglomerates able to increase the stability of their active contents and, consequently, increase their bioavailability.

## Figures and Tables

**Figure 1 foods-11-03860-f001:**
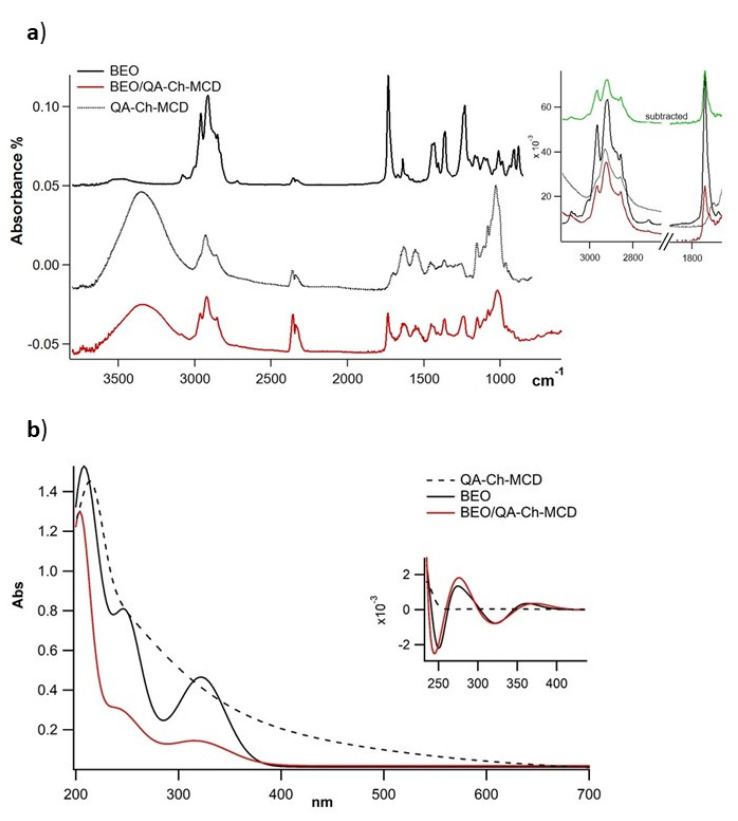
Spectroscopic characterization of BEO (black), BEO/QA-Ch-MCD (red), QA-Ch-MCD (dashed line). (**a**) ATR-FTIR full spectra, and in the box are the overlays of C-H (3000–2800 cm^−1^) and C=O (1700 cm^−1^) stretching bands, the green line represents the spectrum obtained by subtracting QA-Ch-MCD from the BEO/QA-MCD spectrum and evidencing the shift of the 2918 cm^−1^ band. (**b**) UV-Vis full spectra, and in the box are the second derivative profiles evidencing the shift of the 250 nm band.

**Figure 2 foods-11-03860-f002:**
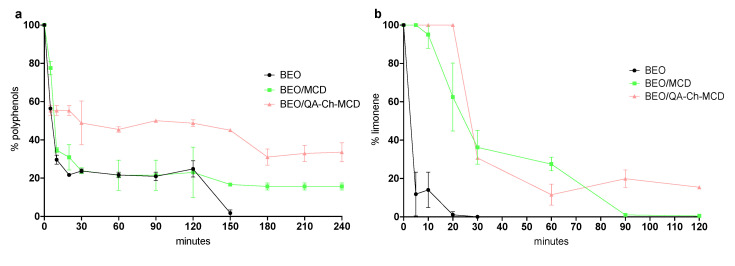
Polyphenol (**a**) or limonene (**b**) stability in SGF under UV irradiation. Means ± SD (*n* = 3).

**Table 1 foods-11-03860-t001:** AE % values for BEO/MCD complexes, obtained by varying the complexation conditions in presence of 10 mg of MCD. Mean ± SD, *n* = 3.

Run	BEO ^1^	Time ^2^	AE %	
	(mL)	(h)	Polyphenols	Limonene
1	200	24	27.7 ± 1.8	27.5 ± 1.5
2	200	48	26.1 ± 2.4	26.1 ± 2.4
3	300	24	18.0 ± 1.0 *	19.5 ± 1.2 *
4	500	24	15.0 ± 1.6 *	14.9 ± 0.9 *

^1^ Amount of BEO in the formulation. ^2^ Incubation time. Data marked by * are significantly different from run 1 (*p* < 0.05).

**Table 2 foods-11-03860-t002:** Diameter distribution and ζ-potential values of BEO/MCD and BEO/QA-Ch-MCD in aqueous solutions. Mean ± SD, *n* = 3.

	Diameter Distribution		ζ-Potential
	Size (nm)	PdI	(mV)
BEO/MCD	359.70 ± 8.50	0.33 ± 0.01	−19.90 ± 0.78
BEO/QA-Ch-MCD	414.20 ± 36.06	0.67 ± 0.06	44.60 ± 5.24

## Data Availability

Data are contained within the article.

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
