# Peer review of "Bergamot Essential Oil: A Method for Introducing It in Solid Dosage Forms"

_foods, 2022, doi:10.3390/foods11233860_

Round 1
Reviewer 1 Report
The manuscript title "Bergamot essential oil: a method for introducing it in solid dosage forms" is basically exploring the Bergamot essential oil in pharmaceutical sciences.
The authors has explored BEO and their complexes to utilze in the solid dosage form development.
The authors does not provide the sufficient methodology, preparation and evaluation of the BEO-cyclodextrin and complexes formation. It is very crucial that the complex has been forming or not and well characterization using molecular dynamic, simulation, DSC, TGA, XRD and so on.
Author only shows the TOC of BGO oil and their Polyphenols and limonene stability under UV irradiation.
Author Response
Point 1: The authors does not provide the sufficient methodology, preparation and evaluation of the BEO-cyclodextrin and complexes formation. It is very crucial that the complex has been forming or not and well characterization using molecular dynamic, simulation, DSC, TGA, XRD and so on.
Response: We introduced a spectroscopic characterization of the complex by ATR-FTIR and UV-visible analysis of the obtained product to evaluate the interaction between BEO and the conjugate QA-Ch-MCD.
Reviewer 2 Report
The communication paper under the title “Bergamot essential oil: a method for introducing it in solid dosage forms” by Zambito et al. describes the preparation of inclusion complex of bergamot essential oil in methyl-β-cyclodextrin and in a quaternary ammonium chitosan derivative coupled with methyl-β-cyclodextrin in order to stabilize and improve the physical and chemical stability of bioactives contained in bergamot essential oil.
In my opinion this communication is well written, have scientific impact and the results are presented clearly.
Author Response
All ok.
Reviewer 3 Report
Dear Authors,
I have included comments on your work in the attachment.
Best Regards

Author Response
Point 1: In my opinion, the "results and discussion" section is too short. Two tables and one figure are a bit too few results. I suggest enriching the work with additional photos, drawings or a table and expand the descriptions.
Response: We separated results from discussion to compare our results with the data already presented in literature, and we added the ATR-FTIR and UV-visible analysis and related figures of the obtained product to evaluate the formation of inclusion complex between BEO and the conjugate QA-Ch-MCD.
Reviewer 4 Report
Dear Authors,
The communication “Bergamot essential oil: a method for introducing it in solid dosage forms” is generally very well written and contains data of some relevance for a general readers as well as of high relevance for specialists in the topic. Although the subject of the paper could be of interest for the readers of the journal, the paper needs some corrections.
Strengths of the paper:
This short communication reports is written concisely and clearly.
Weaknesses of the paper:
I miss even a short discussion of the results. I also believe that a statistical method should be used to demonstrate statistically significant differences.
- Table 1: In order to fine-tune the best complexation conditions statistical treatment should be applied
- Lines: 120-121 - The results presented in Table 1 contradict this statement: “by increasing the amount of BEO in the formulation only a slight increase in AE % was observed”
- Line 130: too much space between words.
- Line 135: “Polydispersity index (PdI)” – there is no information in the “Materials and Methods” about this index.
- Lines: 71, 112, 141: Instead of the word "had" it is better to use the word "was characterized by"
Author Response
Point 1: I miss even a short discussion of the results. I also believe that a statistical method should be used to demonstrate statistically significant differences.
Response: We separated results from discussion to compare the results obtained in our work with the data already presented in literature.
Point 2: Table 1: In order to fine-tune the best complexation conditions statistical treatment should be applied.
Response: The standard deviations are presented in the Table 1 together with the statistical differences.
Point 3: Lines: 120-121 - The results presented in Table 1 contradict this statement: “by increasing the amount of BEO in the formulation only a slight increase in AE % was observed”.
Response: We corrected the statement as suggested.
Point 4: Line 135: “Polydispersity index (PdI)” – there is no information in the “Materials and Methods” about this index.
Response: We introduced the meaning of polydispersity index in the discussion section (line 287).
Point 5: Lines: 71, 112, 141: Instead of the word "had" it is better to use the word "was characterized by"
Response: Corrected (line 153).
Round 2
Reviewer 1 Report
The revised version and responses received by author by adding of the FTIR and UV analysis quit acceptable. Still in my opinion, some DSC, XRD and other data have been required.
Author Response
We agree with the referee that differential Scanning Calorimetry (DSC) is a powerful technique to determine the interaction between guest and host. Generally, it is observed that CDs cavity affects the thermal behaviour of the included guest molecule. The observed effects regard an increased thermal stability (observation of higher degradation temperatures) or a reduced or absence of crystallinity, in most cases. However, in the specific case of BEO/QA-Ch-MCD it is not feasible to acquire any information about the complexation by applying DSC analysis. The reason is that the polymer has a degradation temperature lower than that of BEO and no other transition is present as diagnostic reference temperature. In particular, on the basis of the data already obtained with the conjugate QA-Ch-MD [Piras, A. M. 2018] and with BEO [Navarro-Segura, L. 2019], QA-Ch-MCD has an exothermic peak at ~ 226 °C due to the degradation of quaternary ammonium chitosan whereas BEO has an endothermic peak at ~ 234 °C. Since the polymer degradation is anticipating that of the oil, DSC analysis would be ineffective at detecting an interaction between BEO and polymer. For this reason, other techniques (IR, UV) have been applied to characterize the complex.
Piras, A. M., Zambito, Y., Burgalassi, S., Monti, D., Tampucci, S., Terreni, E., Fabiano, A., Balzano, F., Uccello-Barretta, G., Chetoni, P. A water-soluble, mucoadhesive quaternary ammonium chitosan-methyl-β-cyclodextrin conjugate forming inclusion complexes with dexamethasone. J. Mat. Sci. Mater. Med., 2018, 29(4), 1-13. https://doi.org/10.1007/s10856-018-6048-2
Navarro-Segura, L., Ros-Chumillas, M., López-Cánovas, A. E., García-Ayala, A., López-Gómez, A. Nanoencapsulated essential oils embedded in ice improve the quality and shelf life of fresh whole seabream stored on ice. Heliyon, 2019, 5(6), e01804, https://doi.org/10.1016/j.heliyon.2019.e01804